# Derivatization of Bufadienolides at Carbon-3 of the Steroid Core and Their Consequences for the Interaction with Na^+^,K^+^-ATPase

**DOI:** 10.3390/ijms262211027

**Published:** 2025-11-14

**Authors:** Lucy Kate Ladefoged, Birgit Schiøtt, Natalya U. Fedosova

**Affiliations:** 1Department of Biomedicine, Aarhus University, Høegh-Guldbergsgade 10, 8000 Aarhus C, Denmark; lkl@ravenbiosciences.com; 2Department of Chemistry, Aarhus University, Langelandsgade 140, 8000 Aarhus C, Denmark

**Keywords:** Na^+^,K^+^-ATPase, bufalin, digoxin, bufadienolide, cardenolide, stereochemistry, glycosylation, molecular dynamics

## Abstract

Bufadienolides exert broad-spectrum pharmacological activities relevant to cardiology and novel cancer treatments. Their efficacy, toxicity, and pharmacokinetic profiles are significantly affected by modifications at carbon-3 (C3) of the steroid core. We have applied molecular dynamics simulations to characterize the consequences of (i) variations in size of the substituent at C3, (ii) the type of linker at C3 (ether vs. N-methoxy), and (iii) stereochemistry (C3β vs. C3α) for derivatives’ interactions with Na^+^,K^+^-ATPase. The model compounds included bufalin, bufalin-N-glucose, bufalin-O-glucose as well as digoxigenin, digoxigenin monodigitoxoside and digoxin. It was shown that the optimal size of the substituent is a trade-off between the ability to form stabilizing interactions and steric and entropic interferences. The former is strongly affected by the nature of the linker due to its impact on the spatial position of the ligand: N-methoxy linker imposes rotational restrictions and places the core into a less favorable position compared to an ether bond. Similarly, the change from β- to α-anomer delocalizes the substituent precluding contacts with amino acid residues of the binding site. The presented mechanistic model of bufadienolide interactions with Na^+^,K^+^-ATPase helps to anticipate the consequences of modifications while designing derivatives with high anticancer activity but reduced cardiotoxicity.

## 1. Introduction

Cardiotonic steroids (CTSs) present in both plants and animals have been known for their biological activity for centuries. In Western medicine, CTS isolated from *Digitalis purpurea* have been originally prescribed as diuretics and are still in use for treatment of various heart conditions. Their positive inotropic effect is related to the specific inhibition of the Na^+^,K^+^-ATPase in the heart, but application is limited by a narrow therapeutic index since this enzyme is ubiquitous and crucial for cell homeostasis. In Chinese traditional medicine the spectrum of their application is broad and includes, in addition to cardiology, disorders related to neurology and malignant cell growth. It seems, therefore, that there are many more mechanisms behind cytotoxicity and not all of them anticipate inhibition or even participation of the Na^+^,K^+^-ATPase (for reviews see, e.g., [1,2]). The tumor-suppressing activity in human hepatocellular carcinoma may, for example, involve 14 pathways, since network pharmacological analysis has predicted 82 targets [3]. As a result, several CTS compounds found in toad extract and referred to as Huachansu (from the bufadienolide subfamily of CTS) have been approved by the China Food and Drug Administration (No. Z20050846) for the use in combination with conventional chemotherapy and the extracts from *Nerium oleander* (e.g., Anvirzel and PBI05204) entered Phase I and II clinical trials for their anticancer activity.

In addition to the described biological activities, the CTSs have also been mentioned as anti-viral agents and even made headlines in the United States during the COVID-19 pandemic [4,5,6]. This diversity of pharmacological effects justifies the efforts invested into work on structure–activity relationships (SARs) of the CTS with the hope of diminishing the systemic toxicity and strengthening specific effects by directing compounds toward well-defined single targets, e.g., an isoform of the Na^+^,K^+^-ATPase. The number and diversity of new candidates for drug discovery screening were significantly increased when chemical derivatization schemes were supplemented with microbial/fungal biotransformation approaches applied to known parent CTS molecules [7,8]. These compounds with modified CTS structures were mainly screened for their effects on cell viability while the data on their interactions with the Na^+^,K^+^-ATPase are much more limited. Therefore, in order to separate specific and systemic effects, personalize treatment, and ensure the safety of the patients, it is crucial to untangle the correlations between cell viability, Na^+^,K^+^-ATPase inhibition, and the structures of the compounds. The present paper aims to make beneficial additions to the understanding of the role of stereochemistry, linker type, and size of the substituents in the C3 position of the steroid core (Figure 1) by a combination of molecular modeling and biochemical characterization of the inhibiting potency of chosen CTS towards Na^+^,K^+^-ATPase. The C3 substitution site attracts special attention for several reasons:

First, it has been established that glycosylation or esterification of the C3 position improves solubility, which is of the utmost importance for such hydrophobic compounds as bufalin. Therefore, C3β ester derivatives with different lengths and functional groups were among the newly synthesized compounds aimed to affect viability [9], e.g., a bufalin prodrug with PEG-C3β substituent demonstrated enhanced permeation, circulation time, and improved water solubility [10].

Second, the C3 position of the steroid core is a chiral site, and epimerization is one of the reactions upon biotransformation by microbiota in the human gut affecting absorption and clearance of the drug [11]. As an example, epiforms of bufalin and resibufogenin were identified in rat plasma after oral administration of Huanchansu [12] and epiconfigurations of several bufadienolides were found to exhibit higher and more selective cytotoxicity than the parent compound [13].

Third, natural compounds are O-glycosylated, yet SAR of CTSs toward cancers are often based on series of synthesized MeON-neoglycosides due to the convenience of this modification method [14]. Direct comparison of MeON-neoglycosides and O-glycoside derivatives of digitoxigenin showed similar modes of action, though O-glycosides were more efficient against certain cancer types [15]. These observations, however, have not yet been directly linked to inhibition of Na^+^,K^+^-ATPase.

In short, the C3 position is an excellent spot for CTS derivatization to improve the compounds’ properties as drugs, yet the consequences for binding to Na^+^,K^+^-ATPase remains poorly understood on a molecular level. We characterized the binding of a series of bufadienolides and cardenolides prior to submission of selected CTSs to molecular dynamics (MD) simulations totaling 16 μs of combined simulation time. The resulting trajectories were used to explore each CTS’ binding modes to correlate them with their inhibition patterns.

This paper presents a mechanistic framework describing how and why a substituent at the C3 position of the sterol core affects the binding interactions with Na^+^,K^+^-ATPase. It allows us to predict the outcome of common derivatization strategies and shorten the time and costs of drug design and development.

## 2. Results and Discussion

In order to obtain a more itemized description of the consequences of C3 derivatization for CTS interactions with the Na^+^,K^+^-ATPase, we have chosen arrays based on three bufadienolide aglycones (bufalin, hellebrigenin, and scillarenin) as well as on three cardenolide aglycones (ouabagenin, digoxigenin, and digitoxigenin) for comparison of their binding affinities to Na^+^,K^+^-ATPase (Figure 1A,B). Binding patterns of bufadienolides, and anomers of bufalin derivatives in particular, were evaluated also in the presence of potassium. Typical compounds were subjected to MD simulations. Since affinity of CTSs to Na^+^,K^+^-ATPase depends on the enzyme conformation and is highest towards its phosphorylated states, our choice fell on E2P_i_ phosphoform, a product of back-door phosphorylation with P_i_ as a substrate. It allowed us to perform the binding experiments under equilibrium conditions (and not steady-state as in case of the ATPase reaction) and use the known crystal structures as start points for simulations. In addition, the back-door phosphorylation can occur in the presence of K^+^ [16] as visualized in the crystal structure of bufalin-E2P_i_ complex obtained in the presence of 200 mM KCl [17].

### 2.1. Singly Glycosylated Aglycones Have the Highest Affinity Toward Na^+^,K^+^-ATPase

Data from Figure 1C and Appendix A depict the effect of the size of sugar moiety on the inhibiting properties for three separate arrays of bufadienolides and cardenolides and illustrate the common trend that attachment of a single sugar moiety significantly improves the affinity while further elongation of the chain has a deteriorating effect. The affinities were estimated relative to that of anthroylouabain (AO) due to experimental difficulties and limitations caused by very high affinities yet slow equilibration velocities of some CTSs (see Section 3). The bell-shaped dependencies were observed in both Tris (Figure 1C) and histidine (Appendix A) even though the buffers have varying effects on the affinities of a compound and AO as reflected in their numerical values. We were unable to obtain scillarenin as well as bufalin and ouabagenin diglycosides to complete the series. However, according to data from the literature, scillarenin has an apparent K_D_ of more than double that of proscillaridin A, i.e., singly glycosylated scillarenin [18], and bufalin diglycoside has been found to have a much worse affinity than bufalin-O-glucose [19]. Usage of AO as a proxy for an ouabagenin diglycoside, results in a consistent trend. In short, singly glycosylated cardiotonic steroids have the highest affinity toward Na^+^,K^+^-ATPase relative to both their aglycone and polyglycoside counterparts.

### 2.2. Heterogeneity of Binding and the Role of the Linkage on C3

Heterogeneous binding with the E2P_i_ form of the Na^+^,K^+^-ATPase in the absence of K^+^ has previously been reported for ouabagenin, digitoxigenin, and bufalin [19] and was also observed for all tested CTS aglycons herein. All of them form both fast- and slow-dissociating complexes with the E2P_i_ conformation of the enzyme. K^+^ supports formation of exclusively slow-dissociating complexes. Figure 2A,B illustrate this for hellebrigenin, in line with the data in [17] for bufalin and digitoxigenin. Note, that the K^+^ effect on the affinity of aglycones depends on the subfamily: very small in the case of bufadienolides, but strongly antagonistic for cardenolides.

Glycosylation of CTSs through the natural glycosidic bond results in homogenous binding to Na^+^,K^+^-ATPase regardless of K^+^ (Figure 2A) as previously shown for glycosylated bufalin [19] and cardenolides ouabain and digoxin [17]. Only one of the tested O-linked bufadienolides, scillaren A, continued to display heterogeneous binding (Figure 2C) in the absence of K^+^. Conversely, all analyzed MeON-linked and esterified bufadienolides (bufalin-N-glucose, spin-labeled bufalin and cinobufagin) displayed heterogeneous binding with lower affinities than that of aglycone (Figure 2E, Appendix A, and Refs. [20,21]). In the presence of K^+^, all of them display homogeneous binding as expected. Figure 2D,F illustrates this effect for scillaren A and bufalin-N-glucose.

### 2.3. Stereochemistry of the Substituent on the C3 Position of the Steroid Core

Differential behavior of stereoisomeric steroids has been reported earlier, where anomers of cholesterol varied in their selectivity as well as affinity to target proteins, in addition to differences in their biophysical effects on membranes [22]. Considering the diverse approaches to derivatization of CTS, it would be advantageous to untangle the effect of the stereoconfiguration at C3 on binding to Na^+^,K^+^-ATPase. In the case of bufadienolides, it was shown that 3β-configurations of bufalin and three bufalin derivatives have considerably higher affinity to Na^+^,K^+^-ATPase than their 3α-anomer counterparts [23]. Yet, crystal structures of the Na^+^,K^+^-ATPase with bound bufalin failed to explain the clear selectivity of the enzyme towards configuration with 3β-hydroxyl group over 3α, and the interactions between the substituents on position C3 of the steroid core and the amino acid residues in the CTS binding site are not completely resolved [17,24,25,26].

We evaluated the binding of αC3 isomers of bufalin and bufalin-N-glucose to the Na^+^,K^+^-ATPase in the presence and absence of K^+^ (Figure 3) and confirmed lower affinity for α-isomers. The effect is especially pronounced for bufalin-N-glucose. Additionally, the presence of K^+^ in the binding media had a positive influence on affinity. Although binding kinetics seem to remain heterogeneous for both bufalin and bufalin-N-glucose, the K^+^ effect is more pronounced for α-bufalin (Figure 3).

### 2.4. Molecular Dynamics Simulations of Representatives from Cardenolide and Bufodienolide Subfamilies

As detailed above, the biochemical experiments uncover differences in the inhibiting properties of CTS related to the following:(1)The type of linker at the C3 position of the steroid core,(2)The stereochemistry of the substituent at C3, and(3)The size of substituents at C3.

Yet, the understanding of the intermolecular interactions with the target protein compelling these differences is lacking. In order to explore the underlying mechanisms, we turned to MD simulations. Considering the resource requirements of running MD simulations, we selected one representative from the bufadienolide and cardenolide family. Bufalin was chosen for simulations since it is a known potent drug, and its application is challenged by low water solubility. Attempts to overcome this issue include glycosylation and other types of derivatization at C3. Therefore, bufalin and two bufalin derivatives were included in the study of point 1 and 3: bufalin-O-glucose with a common ether linker, and bufalin-N-glucose with an N-methoxy linker. In the analysis of stereochemical effects (point 2), both anomers of bufalin and bufalin-N-glucose were included due to the immediate relevance to design of drugs with unique pharmacological activity. With the aim to compare bufadienolides with cardenolides in the study of point 3, the array digoxigenin—digoxigenin monodigitoxoside—digoxin was also selected for simulations.

To obtain starting structures for the simulations, each bufadienolide, except bufalin, was flexibly docked into Na^+^,K^+^-ATPase. The starting point for the Na^+^,K^+^-ATPase/bufalin complex was the experimentally solved 4RES structure representing the enzyme in E2P_i_ form co-crystallized with bufalin, i.e., the form functionally characterized in this report [17]. From the docking calculation, the conformation of each bufadienolide with the binding mode most similar to that of the experimentally determined bufalin complex was selected as the starting point for the MD simulations (Appendix A). The starting point for the Na^+^,K^+^-ATPase/digoxin complex was the 4RET crystal structure of that complex [17], and the digoxigenin and digoxigenin monodigitoxoside structures were obtained by sequential removal of sugar moieties/y from the original structure. The complexes were simulated for 500 ns in three repeats (MD1–3). As the Na^+^,K^+^-ATPase/bufalin complex forms the baseline for the analyses, it was simulated in five repeats to improve sampling (see Section 3 for details), data from the Na^+^,K^+^-ATPase/bufalin system has previously been published [27]).

The binding mode, interactions, and stability of each CTS were monitored during the MD simulations. To define binding modes, the steroid core of each CTS was clustered based on its conformation and location within the binding site for all trajectories obtained from the MD simulations. Overall, the clustering revealed four preferred rotational states of the steroid core within the binding site (Figure 4, the rotational axis follows the length of the steroid core). These are referred to as BM0–3 in accordance with our previous paper about bufadienolide/cation interplay [27]. BM0 is equivalent to the binding mode commonly detected in crystal structures of Na^+^,K^+^-ATPase/CTS complexes [17,24,25] and depicts the β-surface of the CTS pointing toward αM2 of Na^+^,K^+^-ATPase. In BM1, BM2, and BM3, the β-surface points toward αM1, αM4, and αM6, respectively (Figure 4A–D).

Table 1 and Appendix A provide an overview of the possible intermolecular hydrogen bonds and their prevalence for individual CTSs in each of the observed binding modes, while Figure 5 displays the lactone–ion II distance distributions. CTS stability within the binding site was evaluated by measuring its movement up and down the site as well as its tilting as reported in Appendix A, respectively. As none of the CTS include acidic nor basic functional groups, the hydrogen bond is the strongest interaction they can form with the protein, and the number and prevalence of hydrogen bonds can therefore serve as a proxy for the enthalpic contribution to binding energy. Estimating the entropic contribution is a more laborious endeavor and has been omitted in the present analyses. However, as restraining multiple degrees of freedom results in higher entropic penalty, it can be assumed that the entropic penalty of retaining a single, specific conformation will be higher when additional glucose moieties are added to the aglycone.

#### 2.4.1. Type of Linker at the C3 Position of the Steroid Core

As N-methoxy glycosylation grew more popular due to the comparable ease of synthesis, the conformational equivalence of N-methoxy and O-links was assessed. Thus, NMR spectroscopy and ab initio modeling revealed slightly different conformational landscapes of disaccharides with glycosidic or N-methoxy linkages [28]. X-ray crystallographic analysis of torsional profiles of digitoxin and its neoglycosylated derivatives also showed small differences for the two linkers. They fell, however, within the range of torsions displayed in the solid state by other cardiac glycosides [29]. Functional studies focused on the cytotoxic properties of ligands while both types of linkers found them almost equally efficient [15]. Langenhan et al., however, observed that cytotoxicity of N-methoxy linked analogs of digitoxin does not correlate with their inhibiting power towards Na^+^,K^+^-ATPase [29]. Thus, even minor differences in conformational landscapes of the molecules due to the linkers might matter enormously for binding to Na^+^,K^+^-ATPase.

The nature of the linkage between the steroid core and the C3 substituent has been chosen as a first step in the analysis of glycosylation in variant forms. Modeling included aglycone bufalin, bufalin-N-glucose, and bufalin-O-glucose. The primary binding mode of the aglycone in the presence of K^+^ is BM1, but BM2 is also observed (69% and 17% of the combined simulation time, respectively, Figure 4E, Table 1) despite simulations being started from BM0 in accord with crystal structures [17,25]. As has been discussed previously, BM0 was not favored in simulations [27]. The other ligands under investigation have their own preferences for binding modes (Table 1, Figure 4). The impact of glycosylation on binding modes is expected since a glucose moiety can form additional interactions with Na^+^,K^+^-ATPase and thus change the preferable binding mode. Yet, the fact that two singly glycosylated bufalin variants behave differently is remarkable as it gives the linker a credit of influence, either due to direct interaction with the protein or to restrictions imposed on glucose rotation relative to the steroid core. As bufalin-N-glucose and bufalin-O-glucose exhibited different inhibition patterns both in the presence and absence of K^+^ (Figure 2E,F, [19]), we have looked for the correlation.

We have previously explored the effect of K^+^ on bufadienolide aglycone binding to Na^+^,K^+^-ATPase using a combination of biochemical experiments and molecular dynamics simulations [27]. In brief, our interpretation of the data was that K^+^ keeps the αM1–4 and αM5–10 bundles in optimal configuration for binding bufadienolides and interacts directly with the bufadienolide lactone. Assuming this hypothesis is true, the inhibition patterns suggest that the presence of a glycosidic bond and at least one glucose moiety can stabilize bufadienolide binding equivalently to K^+^ and in a way that the N-methoxy linked glucose cannot. We therefore simulated and compared the dynamics of bufalin-O-glucose and bufalin-N-glucose with and without K^+^ occupying both cation sites.

Binding of bufalin-N-glucose in the absence of K^+^ is heterogeneous, with a preference for BM0 (81% of the time) stabilized through the core by a hydrogen bond between OH-C14 and T797, and through the sugar moiety by minor interaction with Q111. The lack of support from the cation and only minor stabilizing interactions between the sugar moiety and Q111 render the core-T797 interaction much less prevalent. Bufalin-N-glucose is thus not very stable in the binding site and fluctuates up and down the site (Appendix A) with common tilting of the steroid core (Appendix A). The presence of K^+^ opens for involvement of the lactone ring and the BM0 mode of bufalin-N-glucose becomes near-exclusive (95%) with highly persistent core-T797 and lactone-K^+^ interactions that are supplemented by minor input from hydrogen bonds between glucose to Q111 and T114 (Table 1), in line with the binding kinetics from Figure 2F.

As stated earlier, N-linked substituents did not improve affinity compared to aglycones, and bufalin-N-glucose is not an exception (Figure 2E,F). Even though bufalin and bufalin-N-glucose (both in the presence of K^+^) have very similar interaction patterns (Figure 4E and Figure 5, Table 1, [27]), bufalin binds in BM1 while bufalin-N-glucose occupies the less favorable BM0. Although BM1 potentially opens for an additional contact between the glucose moiety and R880, this particular interaction is difficult to establish. High flexibility of the surface-facing arginine residue as well as the dynamics of the long loop section where R880 is located would hinder the hydrogen bonding and require a considerable loss of entropy, which can rationalize why BM0 is favored by β-bufalin-N-glucose. Preference for BM0 is a reason for the decreased affinity for bufalin-N-glucose compared to bufalin (Figure 2E,F). The overall orientation with the relatively hydrophobic β-surface of bufalin facing more polar αM2 instead of αM1 as in BM1 leads to less favorable vdW interactions. Despite vdW interactions being weak, their combined effect over multiple atoms can alter affinity. A similar drop in affinity due to preference to BM0 was earlier observed for cinobufagin [27].

Bufalin-O-glucose has been reported to form exclusively stable complexes with high affinity in the absence of K^+^ [19]. In the equivalent simulations, the ligand is most often found in BM1 (70%), significantly stabilized by a core-T797 interaction and additional sugar-mediated contacts with E115 and R880 (Table 1, Appendix A). Presence of K^+^ decreases the fraction of BM1 while increasing that of BM2 (now 37% and 58%, respectively). This diminishes input from core-T797 stabilization, and binding is instead supported through ion–lactone interactions. Importantly, in the presence of K^+^, BM1 does not sustain hydrogen bonding between sugar and E115 and R880. According to simulations, K^+^ occupation of site II makes bufalin-O-glucose binding more stable as evidenced by the decrease in ligand tilting (Appendix A) and the lack of movement up and down the binding site (Appendix A). In short, when potassium is not present in site II, bufalin-O-glucose can form sugar–protein interactions in both BM1 and BM2, but when potassium is bound to site II, the sugar interactions are nearly completely lost in favor of the ion–lactone stabilization for both BM1 and BM2. The core-T797 interaction is possible in all of the binding modes, even though glycosylation can place the steroid core at a bigger distance from the cation II site in order to form stabilizing interactions with the protein (Appendix A). These interactions allow to bypass the need for K^+^ to exclusively form slow-dissociating complexes.

It should be stressed that the overall identical binding modes were affected by the presence of glucose and its linkage to the steroid core: bufalin aglycone can sustain a core-T797 hydrogen bond in BM1, but not BM0 nor BM2, while bufalin-N-glucose can sustain it in BM0 and bufalin-O-glucose can in both BM1 and BM2. This leads us to look at the sugar orientations that are influenced by the linker, which propagates down to the increased stabilization of the ligand, and OH-C14 can then form a hydrogen bond to T797.

To assess the correlation between positions of the steroid core and glucose within the binding site, the rotational profiles of the N-methoxy and the commonplace glycosidic bond were analyzed in the context of Na^+^,K^+^-ATPase. The force field parameters for the N-methoxy linker were optimized specifically for these simulations as force field parameters found by analogy could not reproduce the rotation profile obtained by quantum mechanical calculations for the important dihedral angles in question. The parameterization was performed using FFTK [30] and Gaussian [31] as explained in Section 3.

It is expected that the potential energy associated with the dihedral rotation around the C3-O3 and O3-X bonds differs from that of C3-N3 and N3-X bonds, where X is whatever atom is additionally bound to O3 (see Figure 1 for the location of C3 and Appendix A for the N-methoxy linker structure), due to the steric addition by the methyl substituent on the amine linker. The question is, however, whether the change in torsional energy matters in achieving the optimal orientation of both the core and the sugar moiety simultaneously when bound to Na^+^,K^+^-ATPase. We therefore monitored the location of the sugar moiety for each relevant binding mode of the steroid core (Figure 6). Two well-defined sugar orientations are observed for bufalin-N-glucose, which we refer to as site X0 and Y0.

In the absence of K^+^, there is a possibility of hydrogen bonding with Q111 in site Y0 (21% of time spent in BM0, Table 1); however, the core-T797 interaction is lost. Alternatively, the sugar moiety can occupy site X0 unstabilized while allowing for the core-T797 interaction. Interactions with K^+^ shift bufalin-N-glucose almost exclusively to BM0 and allow both glucose contacts to Q111 and T114, and stabilization through the core. Thus, the N-methoxy linker does not fully support the optimal positioning of both the steroid core and the glucose moiety in the absence of K^+^.

Bufalin-O-glucose was found in BM1 and BM2 modes in the absence of K^+^, and simulations again showed two positions of the sugar moiety (Figure 6B,C). The dihedral angles are similar, but the locations are shifted due to the core rotation. Site X1 orients the sugar towards αM6 and is favored in BM1. It supports sugar-mediated interaction with E115 and R880 simultaneously with the core-T797 interaction. Site Y1 is unsupported and is not observed very often. The sites are less defined for bufalin-O-glucose in the less prevalent BM2 (observed for 26% of the time). Site X2 orients the sugar toward αM5-M6 and is stabilized by interaction with R880 only, while no sugar-mediated stabilization is observed for site Y2, and only the core-T797 interaction anchors the ligand. Thus, in the absence on K^+^, both bufalin-N-glucose and bufalin-O-glucose in BM2 have mutually exclusive interactions stabilizing either the core or the sugar moiety, while bufalin-O-glucose in its most prevalent mode (BM1) not only has the possibility of two sugar-mediated interactions, but also displays persistent core-T797 stabilization. Note, it is assumed that the sampling of each binding mode is sufficient for the present analysis as multiple jumps between each rotational state were observed.

#### 2.4.2. The Stereochemistry of the Substituent on C3

Another aspect of glycosylation is the stereochemistry at the C3-attached hydroxyl/glucose moiety. Bufalin and bufalin-N-glucose were used as examples in this endeavor, and affinities for their β-anomers in the absence and presence of K^+^ are reported in Figure 2E,F. The inhibitory potencies of the α-anomers are much, much lower (Figure 3), while crystal structures of the bufalin/Na^+^,K^+^-ATPase complex provides no clue for that dramatic difference [17,25]. The binding modes and behavior of each anomer of bufalin and bufalin-N-glucose were therefore compared to explore these observations.

As previously stated, β-bufalin is very often found in the BM1 mode (69% of the combined simulation time), and visits BM2 to a minor extent. α-bufalin is almost equally found in BM1 and BM2 (39% and 52%, respectively, Figure 4E). The only difference between the two molecular systems is the orientation of the terminal hydroxyl group. For bufalin-N-glucose, the β-isomer strongly favors the BM0 mode (95% of time), while the α-isomer favors BM1 (92% of time, Figure 4F). Once again, each compound favors a different binding mode.

β-bufalin was found to directly interact with K^+^ in cation site II in both BM1 and BM2. Its core forms a hydrogen bond between the hydroxyl group at C14 (Figure 1A) and T797 in over 90% of simulation time in BM1 while no such support is found in BM2 (Table 1). On the contrary, the terminal hydroxyl group does not form polar interactions in BM1, but in BM2 it can interact with either Q111 or N122 (Table 1). Note that the X-ray crystal structures of Na^+^,K^+^-ATPase/bufalin complexes place bufalin in BM0 in accord with other CTSs [17,25], which might be a result of more favorable crystal contacts. The same interaction pattern, i.e., direct interaction with the K^+^ in site II (Figure 5) and hydrogen bonding between OH-C14 and T797 were detected for the steroid core of α-bufalin in BM1, but the terminal hydroxyl group could no longer interact with the protein in BM2 (Table 1). Simulations without ions in cation site I and II revealed that β-bufalin no longer visits the BM2 mode [27]. The lack of stabilization from the ion is even more deleterious for α-bufalin since it also lacks the ability to interact with Q111 and N122 through OH-C3.

β-bufalin-N-glucose is stabilized by multiple polar interactions in its favored mode, BM0. To briefly recapitulate, the core interacts with T797, while the sugar moiety can interact with Q111 or T114, and the lactone coordinates K^+^ directly. On the other hand, α-bufalin-N-glucose favors BM1 (91%), which is stabilized by core interaction with T797 (Figure 4, Table 1) and by direct interaction with K^+^ (Figure 5). The sugar moiety, however, does not partake in any interactions with the protein. α-bufalin-N-glucose thus forms less stabilizing contacts with the protein and therefore has lower affinity. The change from β- to α-isomer relocalizes the hydroxyl or sugar moiety, thus, preventing additional contacts with the protein as revealed by simulations. This is analogous to the observations for the effect of linker type; however, shifting between anomers perturbs the conformational landscape to a greater extent and thus leads to more pronounced effects on affinity.

#### 2.4.3. A Bell-Shaped Dependence on the Size of Substituents at C3

The last issue that needs consideration in connection with derivatization of the C3 position is the size of the substituent. Despite differences in both steroid cores and sugar moieties in the tested CTS, the singly glycosylated molecules were consistently the most affine (Figure 1C, Appendix A). The benefit of glycosylation over the aglycone is anticipated since the number of hydroxyl groups capable of bonding is increased. Further elongation of the sugar moiety, however, has a negative effect on affinity, despite supplementation of hydroxyl groups. We therefore looked at the consequences of elongation of sugar moiety for binding to Na^+^,K^+^-ATPase.

The above analysis of the type of linkage at C3 and the effect of stereochemistry together with the previously published results [27] provides a sound platform for the understanding of binding behavior of bufalin. Briefly speaking, the data revealed that the sugar moiety of bufalin-O-glucose forms hydrogen bonds with the protein more often than the terminal hydroxyl group of bufalin, thus stabilizing the ligand in the site. These interactions become more important in the absence of K^+^ (no lactone-cation interactions) as seen in simulations of bufalin-O-glucose and reflected in its monophasic inhibition pattern [19].

In accord with the findings herein, spin labeled cinobufagin and bufalin consistently showed heterogeneity of binding and lower affinity compared to parent compounds [20,21]. Although of near-identical size, these derivatives (with 5- or 6-membered nitroxide rings attached at C3) were not stabilized by these substitutions since the spin-labels did not have the hydroxyl groups necessary for stabilizing interactions. Not counting the linker, the spin-labels only have a single radical oxygen to partake in hydrogen bonds, but this oxygen points toward the solvent instead of the loop regions when bound to Na^+^,K^+^-ATPase [20,21]. The linkers’ ketones may only serve as hydrogen bond acceptors.

Interpretation of the glycosylation effects requires caution, since derivatization also results in changes in physical properties of the substances, and may affect affinity estimations. We therefore evaluated the solubility (S) and lipophilicity (P) of each CTS using ALOGPS 2.1 [32] (Appendix A). According to the calculated estimates, bufalin-O-glucose and bufalin-N-glucose are not very different and therefore the conclusions made on the linker effect are fair. In contrast, bufalin has the lowest water solubility of all CTSs and the highest lipophilicity. Therefore, its affinity evaluation might be biased due to bufalin partition into the lipid bilayer and overestimation of the effective concentration in the water phase.

Therefore, we turned to a digoxigenin-based array and modeled digoxigenin, digoxigenin monodigitoxoside, and digoxin, which also allows us to explore a cardenolide as a test system. The estimated solubility and lipophilicity of the three molecules are highly similar (Appendix A).

The simulations revealed that none of them bind directly to the Mg^2+^ ion in ion site II (Figure 5), and that the binding mode of digoxigenin is least persistent and shifts between BM0, BM1, and BM2 (28%, 43%, and 20% of combined simulation time, respectively, Figure 4H). Adding one sugar moiety shifts the balance between BM0 and BM1 while abandoning BM2 entirely (55%, 44%, and 0%, respectively), and adding three sugar moieties shifts the balance even further toward BM0 (79% time spent in BM0 and 19% of time spent in BM1) in accord with crystal structures [17,25]. The interaction analysis (Table 1) reveals overall consistent interactions of the steroid core in BM0 and BM1 for digoxigenin, digoxigenin monodigitoxoside, and digoxin, but a single glucose moiety allows for an additional hydrogen bond to E312, which was not observed for digoxin with three sugar moieties. The same amino acid is involved in contact between rhamnose in the ouabain-Na^+^,K^+^-ATPase complex [25]. Closer inspection revealed that the hydrogen bond donor for the observed interaction was the hydroxyl group of the 4th glucose carbon. This oxygen atom is part of the 1,4-glucoside bond in digoxin and can therefore no longer act as a hydrogen bond donor. The addition of two more glucose units when comparing digoxigenin monodigitoxoside and digoxin also imposes steric restrictions that make it difficult to encounter E312 and to form a hydrogen bond—even if the glutamate side chain should become protonated. These steric restrictions of digoxin also influence the BM1 mode as reflected by the absence of interaction between the steroid core and N122 observed for digoxigenin and digoxigenin monodigitoxoside. Finally, even though we have not explored the entropic penalties directly, it is evident that the reduction in dynamic range for the longer sugar chains is larger than for the shorter chains (Appendix A). Interestingly, the dynamic range was observed to be markedly more restrained in BM1 compared to BM0 for both digoxigenin monodigitoxoside and digoxin. In short, the affinity of digoxigenin monodigitoxoside is higher than the affinity of both digoxigenin and digoxin due to the number of stabilizing interactions combined with minimal steric and entropic restrictions.

Katz et al. [33] explored a range of alkyl moieties substituting the distal sugar moiety of digoxin and detected drastic changes in selectivity towards isoforms of Na^+^,K^+^-ATPase. The digoxin derivative with the highest affinity toward the α1β1γ isoform (the complex of interest in the context of this report) had 3,3-dimethylcyclobutylmethanamine perhydro-1,4-oxazepine instead of the third sugar moiety. This substitution leaves almost no possibility for hydrogen bonding with the Na^+^,K^+^-ATPase through the distal moiety. The authors argue that the hydrophobic moiety likely interacts with W894 and possibly near Q84 of the β1 subunit due to the ligand’s isoform selectivity profile. Assuming the binding orientation of the modified digoxin is similar to that of digoxin, this explanation is reasonable. In both BM0 and BM1, the third sugar moiety of digoxin was observed to be in this location. However, unlike in α1β3γ, this region holds many polar residues in α1β1γ, and modified digoxin displayed improved affinity for both isoforms. According to the simulations of digoxin in BM0, the far most common binding mode, the sugar moieties are hugely flexible and in the most bent conformation the third sugar moiety reaches the lipid headgroups of the surrounding bilayer (Appendix A). This conformation would be highly agreeable with the modified digoxin presented by Katz et al. and may pose an alternative explanation for the improvement in affinity across isoforms.

## 3. Materials and Methods

### 3.1. Biochemical Characterization of E2P_i_-CTS Complexes

Enzyme preparation. Purified pig kidney Na^+^,K^+^-ATPase was prepared as previously described [34]. The specific ATPase activity of the preparation was about 1800 µmol P_i_/hour per mg protein at 37 °C.

Binding of CTS to the E2P_i_ conformational state of Na^+^,K^+^-ATPase. The effect of cations on CTS interactions with E2P_i_ state was estimated from the residual Na^+^,K^+^-ATPase activity after preincubation of the individual CTS with the enzyme essentially as described by Laursen et al. [17]. The data were analyzed as in Yatime et al. [35] using freeware KyPlot 5 (https://www.kyenslab.com/en-us/, accessed on 11 November 2025).

Evaluation of relative affinities for CTS interactions with the E2P_i_. Specific affinities of the Na^+^,K^+^-ATPase in the E2P_i_ conformation towards individual CTS were estimated relative to that towards AO. The experiments were performed on a SPEX Fluorolog-3 spectrofluorometer equipped with a thermostated cell compartment. Due to high affinities though low equilibrating velocities of interactions, the enzyme (0.1 µM) was incubated overnight with the equimolar concentrations of AO and the CTS under investigation (0.2 µM/0.2 µM) in the standard conditions (Tris 40 mM, pH 7.4, inorganic phosphate 3 mM, MgCl_2_ 3 mM). The fluorescence (excitation 370 nm, emission 485 nm) was measured and compared with the fluorescence response from 0.2 µM AO alone. The fluorescence from the sample containing enzyme in the presence of 0.2 µM AO and 1 mM ouabain had been used as a reference for free AO response and subtracted from all values. The ratio of the residual AO fluorescence to the decrease in its fluorescence induced by the presence of another CTS (X), added in equal concentration, is proportional to (AO/K_AO_)/(X/K_X_) and reflects their relative affinities. For example, 50% decrease in AOfluorescence induced by a compound implies their equal affinities or equal equilibrium dissociation constants). The kinetics of interactions between AO and E2P_i_ under identical conditions was assessed in Fruergaard et al. [36]. Figure 2 in that paper shows: time-dependent changes in AO fluorescence as a function of its concentration (binding velocities and amplitudes), calculated second-order association rate constants and a time course of AO dissociation from E2P_i_. The absolute affinity was estimated to be higher than 0.1 nM.

### 3.2. Computational Protein and Ligand Preparation

The protein model used for Na^+^,K^+^-ATPase/bufadienolide complexes was prepared from PDB entry 4RES [17] and the protein model used for Na^+^,K^+^-ATPase/cardenolide complexes was prepared from PDB entry 4RET [19] (both α1β1γ subunits from pig) using the Protein Preparation Wizard [37] in Maestro (Schrödinger Suite 2019, Schrödinger LLC, New York, NY, USA). Chains A, B, and G from 4RET were used as these chains had fewest missing residues, and three water molecules coordinating Mg^2+^ were retained during preparation and included in the following simulations. Preparation entailed capping of termini, addition of hydrogen atoms, potential flipping of His, Gln, and Asn, and a restrained minimization of the protein (max. 0.3 Å RMSD for heavy atoms). The protonation states of titratable residues were assessed by PROPKA 3 [38]; however, the protonation states of residues in the ion binding site were manually adjusted to be in accord with experiments by the Roux lab [39,40]. The resulting models included disulfide bridges between CysB126-CysB149, CysB159-CysB175, and CysB213-CysB276; neutral AspA808, AspA926, GluA244, GluA327, GluA779, and GluA954; histidines HisB212, HisA286, HisA383 (only 4RET-based structure), HisA517, HisA550, HisA613, HisA659, HisA678, HisA875, and HisA912 were modeled as the ε-tautomer; and AspA369 was phosphorylated, while all other residues were modeled in the default state. The parameters for the phosphorylated aspartic acid were taken from Damjanovic et al. [41].

The chemical structure of β-bufalin was extracted from PDB entry 4RES [17], and α-bufalin, β-bufalin-N-glucose, α-bufalin-N-glucose, and β-bufalin-O-glucose (Figure 1) were adjusted manually from the bufalin structure using the build panel available in Maestro (Schrödinger Suite 2019). Similarly, the digoxin structure was extracted from PDB entry 4RET [17], while digoxigenin and digoxin monodigitoxoside were built from the digoxin structure. All compounds were minimized using a conjugate gradient algorithm in 5000 steps and submitted to a conformational search by using a mixed torsional/low mode sampling algorithm as implemented in MacroModel (Schrödinger Suite 2019). The lowest energy conformation of each compound was used in the docking calculation and for force field parameter generation.

### 3.3. Docking Calculations

All docking calculations were performed using the induced fit docking protocol [42] employing Glide and Prime (Schrödinger Suite 2019). In the initial docking all vdW interactions were scaled to 50%, the SP level was applied, and a maximum of 200 poses were allowed. The centroid of the binding site was defined based on the co-crystallized bufalin (PDB ID: 4RES [17]). In the optimization step, residues within 5 Å of the ligand were subjected to side chain optimization. The final docking step was performed in XP, and a maximum of 100 poses with associated energies within 30 kcal/mol of the lowest energy pose were reported in the results. The resulting poses were clustered based on their in-place conformation using the conformer cluster script available in Maestro (Schrödinger Suite 2019), and the pose from the cluster closest resembling the co-crystallized bufalin molecule with the lowest RMSD to the co-crystallized bufalin was used as starting point for the MD simulations. Ion transport sites I and II were occupied by K^+^ ions in the calculations.

### 3.4. System Building for MD Simulations

Each simulated system contains one Na^+^,K^+^-ATPase, one cardiotonic steroid, 0–2 structural cations, a POPC membrane patch, solvent and 0.2 M KCl according to Table 2.

The systems were built from scratch using a combined coarse grain/atomistic (CG/AA) approach as outlined below for the case of the Na^+^,K^+^-ATPase/bufalin/2xK^+^ system, while in the remaining systems containing bufadienolides, the protein with bound ligand and structural ions were simply swapped into the first atomistic system before being equilibrated (Table 3). Similarly, the Na^+^,K^+^-ATPase/bufalin-N-glucose/2xK^+^ and Na^+^,K^+^-ATPase/digoxin/Mg^2+^ systems were built from scratch using the CG/AA approach, while the remaining glycosylated bufadienolide and cardenolide systems were built by swapping protein, ligand, and ion into the backmapped membrane system and equilibrated independently in AA.

The Na^+^,K^+^-ATPase/bufalin/2xK^+^ system was aligned onto the 4RES OPM [43] structure to center the protein in coordinate space. A CG POPC membrane was then built around the protein in the xy plane using Insane [44] and Martinize tools available from the Marrink group’s homepage [45]. The system was solvated and neutralized with 0.2 M KCl before being minimized and equilibrated according to step 1 and 2 in Table 3. The system was then converted into its atomistic equivalent using the Backward tool [46]. In order to ensure that the starting point of the AA simulations was exactly as intended during protein preparation, the backmapped protein was replaced by the original Na^+^,K^+^-ATPase/ligand/ion complex. The atomistic system was then minimized using a conjugate gradient algorithm and further equilibrated in AA resolution before production runs, as outlined in step 3 to 5 in Table 3. As we are interested in the ligand dynamics and stability within the protein, we perform data analysis immediately following release of position restraints. Each system repeat was equilibrated separately to ensure maximal sampling of each molecular system. Simulations of Na^+^,K^+^-ATPase/ligand complexes without structural ions were started from the same conformation as the equivalent systems that included the ions as opposed to the conformations detected in the IFD calculations. In all simulations, the RMSD of C_α_ atoms was observed to converge almost immediately (Appendix A).

### 3.5. Molecular Dynamics Simulations

All simulations were performed in Gromacs 2019.2 using periodic boundary conditions. For the CG simulations, the MARTINI 2.2 force field was used [47,48,49]. The vdW interactions were treated using cut-offs at 11 Å and the potential-shift-Verlet modifier, while electrostatic interactions were treated using the reaction-field method cut-off at 11 Å and a dielectric constant of 0 (=infinite) beyond the cut-off. The neighbor list was maintained using Verlet buffer lists. Temperature was kept at 310 K and pressure at 1 bar. The AA resolution simulations were performed using the CHARMM36m force field [50,51,52], the TIPS3P water model [53], and CHARMM-compatible ligand parameters as described below. The vdW interactions were treated by cut-offs at 12 Å and a force-switch modifier after 10 Å, while electrostatic interactions were treated using PME. The neighbor list was maintained using Verlet buffer lists, and bonds linking hydrogen atoms to heavy atoms were restrained using LINCS [54]. The temperature was maintained at 310 K using a coupling constant of 1, and pressure was maintained at 1 bar using a coupling constant of 4, a compressibility factor of 4.5 × 10^−5^, and semiisotropic coupling to xy and z dimensions separately.

### 3.6. Force Field Parameters for Ligands and Phosphorylated Aspartate

Force field parameters for bufalin were obtained by analogy using the ParamChem webserver and the CHARMM generalized force field (CGenFF) [55] as reported by Ladefoged et al. [27]. Parameters for bufalin-O-glucose were constructed using the above bufalin parameters and the CHARMM carbohydrate parameters [56,57]. Parameters for bufalin-N-glucose were constructed equivalently; however, the special N-OMe linker was parameterized from scratch (see below) as no existing parameters could reproduce the expected linker behavior (rotational energy scan performed at the MP2 level of theory, including the parameters obtained for N-linked glycosylation by Guvench et al. [57]). The parameters for digoxin and digoxin monodigitoxoside were based on CGenFF and CHARMM carbohydrate parameters [55,56]. The CGenFF parameters were obtained from the ParamChem webserver, and the associated penalties were assessed and indicated the parameters were a good fit. In order to retain the correct stereochemistry surrounding the O-link between the steroid core and the first carbohydrate, the dihedral parameters in ring A of the steroid core were taken from the CHARMM carbohydrate force field. The digitoxigenin parameters were identical to the core parameters applied for digoxin and digoxin monodigitoxoside. The applied parameters can be found in Appendix A.

In order to accurately sample the three rotatable bonds in the N-OMe linker in bufalin-N-glucose, five dihedral angles had to be parameterized from scratch (C_sugar_-N_linker_-O_linker_-C_linker_, C_core_-C_core_-N_linker_-C_sugar_, C_core_-C_core_-N_linker_-O_linker_, O_sugar_-C_sugar_-N_linker_-C_core_, and O_sugar_-C_sugar_-N_linker_-O_linker_). The parameters were constructed using the Force Field Toolkit [30] in combination with Gaussian16 [31] and NAMD 2.11 [58] based on the test-molecule shown in Figure 7A. Rotational energy scans were obtained for each dihedral angle (±180° in 10° increments using MP2 and the 6-31g(d) basis set in order to be compatible with the CHARMM force field) and the force field parameters were iteratively fitted. The final fit had a root mean squared error of 1.092 kcal/mol and reproduced the energy landscape well (Figure 7B). The resulting parameters can be found in Appendix A.

All ligand parameters were validated based on short 10 ns simulations of each ligand in explicit solvent. The stereochemistry of the C3 atom linking the steroid core and the hydroxyl group or carbohydrate as well as the stereochemistry surrounding the 1-4 inter-carbohydrate links were stable, and the carbohydrates were consistently in appropriate chair conformations. In the case of the N-linker, the obtained parameters do not interfere with the appropriate behavior of the steroid core and sugar moiety, the linker bonds retain their α- or β-configuration throughout the simulations as expected, and the N-OMe bond samples appropriate minima.

The CHARMM compatible parameters for the phosphorylated aspartate were taken from Damjanovic et al. [41].

### 3.7. Computational Analyses

All analyses were performed using in-house scripts. All figures of molecular systems were made using VMD 1.9.3 [59] or ChemDraw Pro 17.0.

## 4. Conclusions

The present paper addresses the effects of modifications of cardiotonic steroids (the nature of the linker at C3, its stereo configuration, and the size of the substituent) on their interactions with the Na^+^,K^+^-ATPase. Application of biochemical methods in tandem with computational studies of representative bufadienolide and cardenolide compounds have led to the following conclusions:(1)Glycosidic and neo-glycosidic linkages between the steroid core and the first sugar moiety are not equivalent due to small, but important differences in torsional profiles and therefore differences in interaction patterns with the Na^+^,K^+^-ATPase;(2)The chirality at the C3 position greatly influences binding to Na^+^,K^+^-ATPase as the α-anomer orients the sugar moiety toward the solvent instead of toward the protein, where it would ordinarily form stabilizing hydrogen bonds;(3)Singly glycosylated aglycones are most affine due to the increased number of stabilizing interactions combined with minimal steric and entropic restrictions.

The ability to predict the consequences of different modifications of CTSs for their interactions with Na^+^,K^+^-ATPase is paramount for the rapid development of novel drugs based on the known natural precursors. CTSs are currently used in treatment of heart failure, but also intensely screened for their anticancer effects. In the pursuit of developing CTSs into novel drugs, N-methoxy-neoglycosides have become popular due to the relative ease of synthesis and the assumed equivalent binding compared to cardiotonic glycosides. However, as described above, the linkers affect kinetics of interactions and might play a significant role in the optimization process. Another challenge in the transition from the bench to the bedside and in personalization of treatment lies in the susceptibility of CTSs to biotransformation. The human intestinal microbiome participates in the CTS metabolism and increases the number of pharmacological compounds by, e.g., epimerization or oxidation [60]. The consequences of epimerization have been described here for bufadienolides. To summarize, the mechanistic framework based on converging results from molecular dynamic simulations and functional characterization will assist in the development of more effective and safer treatments, especially in oncology, by minimizing the inherent drawbacks of bufadienolides while keeping their potent biological activity.

## Figures and Tables

**Figure 1 ijms-26-11027-f001:**
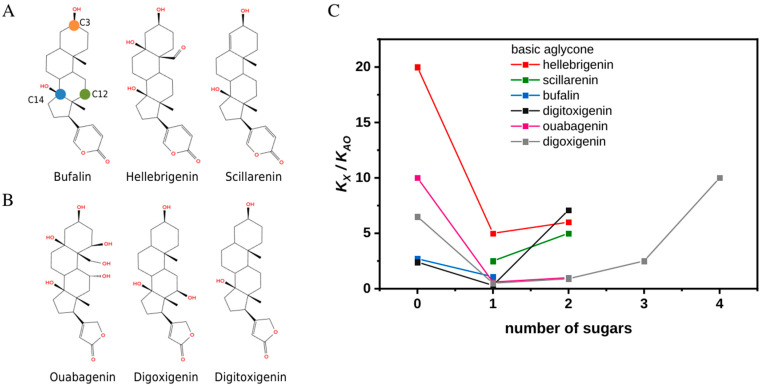
Structures and affinities of selected cardiotonic steroids toward Na^+^,K^+^-ATPase in the E2P_i_ conformation. (**A**) The chemical structures of bufalin, hellebrigenin, and scillarenin from left to right. The location of atoms C3, C12, and C14 (orange, green, and blue circles, respectively) are shown on bufalin. (**B**) Chemical structures of ouabagenin, digoxigenin, and digitoxigenin from left to right. (**C**) The ratio of the dissociation constants of CTS (K_X_) to that of anthroylouabain (AO) (K_AO_) as a function of a degree of glycosylation. Experiments were performed in Tris buffer and in the absence of K^+^. All compounds are O-glycosylated. See Appendix A for complete chemical structures of the compounds with the highest number of sugar residues.

**Figure 2 ijms-26-11027-f002:**
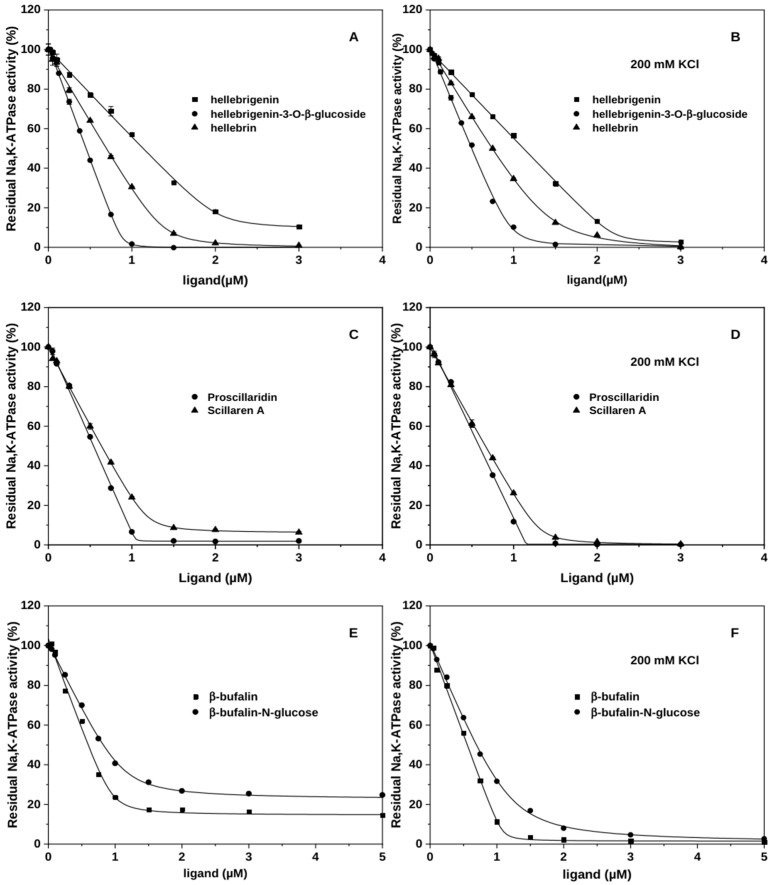
Interactions of bufadienolides with Na^+^,K^+^-ATPase, effects of glycosylation and K^+^. (**A**–**F**) The amount of CTS-bound enzyme is reflected in the percentage of inhibition of the enzyme activity under optimal conditions. Bufadienolides have been incubated with the E2P_i_ conformation of the Na^+^,K^+^-ATPase in the absence of additional cations (**A**,**C**,**E**) or in the presence of 200 mM KCl (**B**,**D**,**F**) prior to activity measurements. The data were fitted to a square-root equation as described in [17].

**Figure 3 ijms-26-11027-f003:**
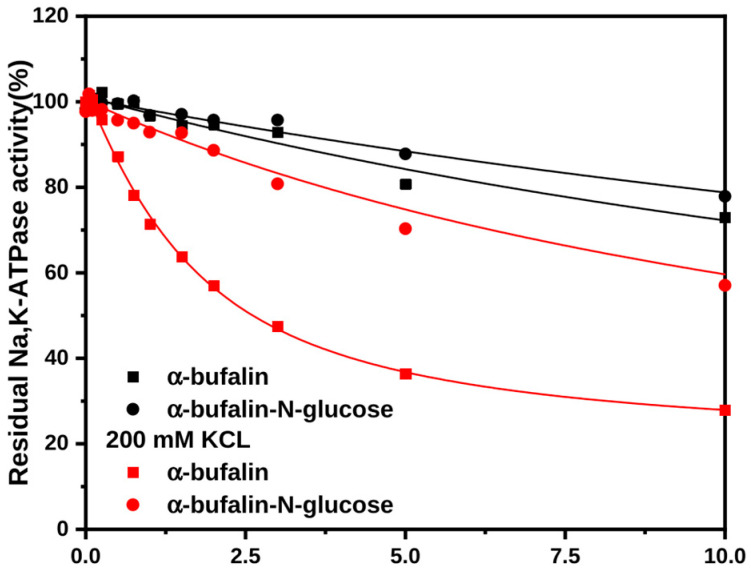
Interactions of αC3 anomers of bufalin and bufalin-N-glucose with the Na^+^,K^+^-ATPase in the presence and absence of K^+^. The amount of CTS-bound enzyme was estimated from the percentage of inhibition of the enzyme activity under optimal conditions. Bufadienolides have been incubated with the E2P_i_ conformation of the Na^+^,K^+^-ATPase in the absence or in the presence of 200 mM KCl prior to activity measurements. The data were fitted to a square-root equation as described in [17].

**Figure 4 ijms-26-11027-f004:**
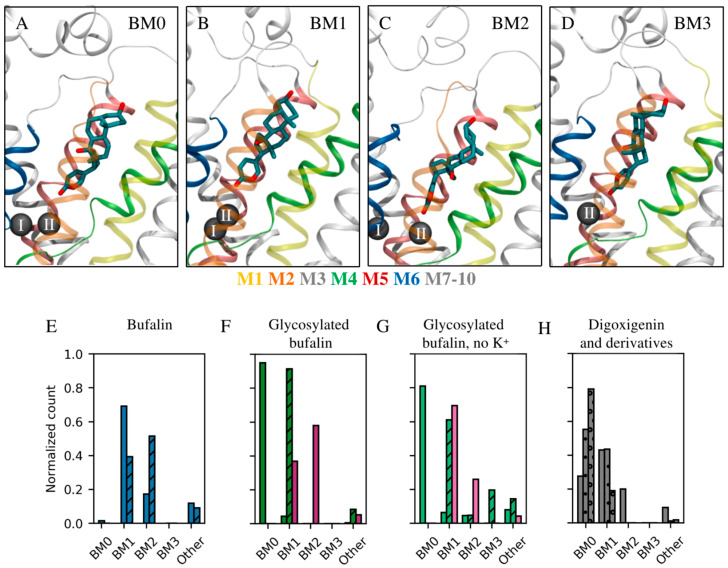
Common binding modes. (**A**–**D**) The four most common binding modes detected as illustrated by β-bufalin in Na^+^,K^+^-ATPase with helices shown in color as indicated below (**A**–**D**). (**A**) BM0 in which the methyl groups on the steroid core point toward αM2 similarly to crystallographic observations; (**B**) BM1 in which the methyl groups point toward αM1; (**C**) BM2 in which the methyl groups point toward αM4; (**D**) BM3 in which the methyl groups point toward αM6. Na^+^,K^+^-ATPase is shown in ribbons, while the ions in site I and II are indicated by black spheres. αM1 and αM5 are transparent so as to not obstruct the view of the ligand. (**E**–**H**) The prevalence of each binding mode for each ligand relative to the simulation time. (**E**) Bufalin is shown in blue with the β- and α-stereoisomer in plain and striped, respectively. (**F**,**G**) Bufalin-O-glucose (magenta) and bufalin-N-glucose (β-form (plain), α-form (striped)). Panel (**G**) depicts the binding mode prevalence with no structural ions present in the MD simulations (lighter hues). (**H**) Digoxigenin (plain), digoxin monodigitoxoside (dotted), and digoxin (circled).

**Figure 5 ijms-26-11027-f005:**
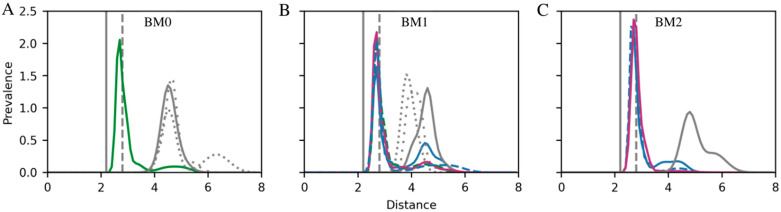
Interaction distance between lactone and the ion in cation site II for (**A**) BM0, (**B**) BM1, and (**C**) BM2. The bufadienolides bufalin (blue), bufalin-N-glucose (green), and bufalin-O-glucose (magenta) are shown with β-isomers as solid lines and α-isomers as dashed lines. Cardenolides are all shown in gray: digoxigenin as a solid line, digoxin monodigitoxoside as a tightly dotted line, and digoxin as a loosely dotted line. Each distribution was calculated using a Gaussian Mixture model with the number of Gaussian components determined using the Bayesian information criterion. BM3 was not detected in simulations of Na^+^,K^+^-ATPase with occupied cation sites. Note that bufadienolides were simulated with bound K^+^ which has an optimal coordination distance of 2.8 Å (dashed gray line), while cardenolides were simulated with bound Mg^2+^ which has a coordination distance of 2.1 Å (gray line). Only binding modes that are observed for more than 5% of the total simulation time are included.

**Figure 6 ijms-26-11027-f006:**
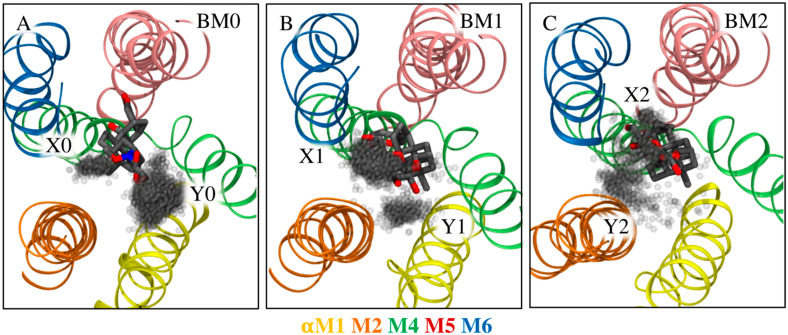
Orientations of sugar moieties relative to common binding modes of (**A**) bufalin-N-glucose and (**B**,**C**) bufalin-O-glucose in simulations without K^+^ atoms. The CTS is viewed from the extracellular environment, perpendicular to the membrane. Each gray dot is the location of the oxygen atom of the *para*-hydroxyl group. Each clustered site is named X and Y as indicated in the figure. All trajectories are aligned on αM1–6, and only frames representative of a given binding mode are included in each image.

**Figure 7 ijms-26-11027-f007:**
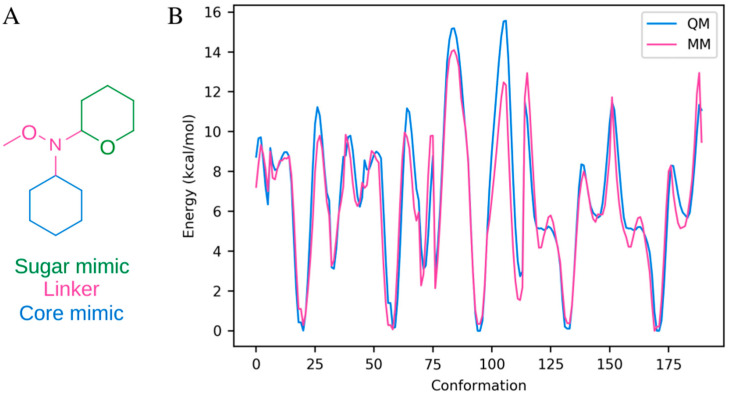
Linker parameterization. (**A**) The molecule used for parameterization. (**B**) The quantum mechanical (QM) energy profile along with the energy profile resulting from the fitted parameters (MM). The energy profile was constructed from five rotational profiles from the molecule in panel (**A**) (C_sugar_-N_linker_-O_linker_-C_linker_, C_core_-C_core_-N_linker_-C_sugar_, C_core_-C_core_-N_linker_-O_linker_, O_sugar_-C_sugar_-N_linker_-C_core_, and O_sugar_-C_sugar_-N_linker_-O_linker_).

**Table 1 ijms-26-11027-t001:** Heatmap of intermolecular interaction prevalence for each CTS in each of its observed binding modes. The prevalence of binding modes is also displayed. Only binding modes that are observed for more than 5% of the total simulation time are included. The BM prevalence indicates the percentage of time a particular CTS spends in a specific binding mode, and the fractions in the interaction matrix denote the prevalence of an interaction for that particular binding mode only. Each cell is colored according to the interaction prevalence, with 0% being white and 100% being green.

			Core Interaction						Hydroxyl at C3/Sugar Interaction							
			Hydroxyl at C14			Hydroxyl at C12										
CTS	BM Prevalence	BM	T797	I315	N122	T797	I315	N122	Q111	T114	E115	N120	D121	N122	E312	R880
β-Bufalin	69%	BM1	91%	0%	7%				0%	0%	0%	0%	0%	0%	0%	6%
	17%	BM2	0%	0%	3%				36%	0%	0%	0%	0%	39%	0%	0%
α-Bufalin	39%	BM1	74%	0%	10%				0%	0%	0%	0%	0%	0%	0%	0%
	52%	BM2	0%	0%	0%				2%	0%	0%	0%	1%	3%	0%	0%
β-Bufalin-N-glucose	95%	BM0	95%	0%	0%				14%	13%	0%	0%	0%	0%	1%	0%
β-Bufalin-N-glucose (no ions)	81%	BM0	57%	0%	5%				21%	0%	0%	1%	0%	0%	2%	3%
	6%	BM1	0%	0%	62%				0%	0%	0%	0%	0%	0%	0%	28%
	5%	BM2	1%	0%	0%				0%	0%	0%	0%	0%	0%	1%	0%
α-Bufalin-N-glucose	91%	BM1	67%	0%	11%				0%	0%	0%	0%	2%	0%	0%	0%
α-Bufalin-N-glucose (no ions)	61%	BM1	18%	0%	2%				3%	0%	1%	0%	0%	0%	0%	0%
	5%	BM2	0%	0%	88%				0%	0%	6%	0%	0%	0%	0%	0%
	20%	BM3	0%	0%	0%				0%	0%	0%	0%	0%	0%	0%	3%
β-Bufalin-O-glucose	37%	BM1	90%	0%	9%				0%	0%	0%	0%	0%	0%	0%	3%
	58%	BM2	39%	0%	2%				1%	6%	6%	0%	0%	0%	2%	0%
β-Bufalin-O-glucose (no ions)	70%	BM1	88%	0%	1%				0%	0%	19%	0%	0%	0%	4%	21%
	26%	BM2	9%	0%	1%				4%	0%	0%	0%	0%	0%	5%	26%
Digoxigenin	28%	BM0	71%	0%	0%	0%	0%	57%	0%	0%	0%	0%	0%	0%	0%	11%
	43%	BM1	69%	0%	20%	0%	33%	1%	0%	0%	0%	0%	0%	0%	0%	3%
	20%	BM2	0%	0%	2%	5%	0%	0%	28%	0%	0%	6%	0%	0%	0%	0%
Digoxigenin monodigitoxoside	55%	BM0	82%	0%	0%	0%	0%	82%	0%	0%	0%	0%	0%	0%	34%	6%
	44%	BM1	83%	0%	18%	0%	0%	0%	0%	0%	0%	0%	0%	0%	2%	16%
Digoxin	79%	BM0	78%	0%	0%	0%	0%	68%	0%	0%	0%	0%	0%	0%	0%	0%
	19%	BM1	82%	0%	0%	0%	0%	0%	0%	0%	0%	0%	0%	0%	0%	0%

**Table 2 ijms-26-11027-t002:** Overview of simulated molecular systems.

Bound Steroid	Stereochemistry at C3	Structural Cations	Simulation Length (ns)	n Repeat Simulations
Bufalin	β	2xK^+^	500	5
Bufalin	α	2xK^+^	500	3
Bufalin-N-glucose	β	2xK^+^	500	3
Bufalin-N-glucose	β	none	500	3
Bufalin-N-glucose	α	2xK^+^	500	3
Bufalin-O-glucose	β	2xK^+^	500	3
Bufalin-O-glucose	β	none	500	3
Digoxigenin	β	1xMg^2+^	500	3
Digoxin monodigitoxoside	β	1xMg^2+^	500	3
Digoxin	β	1xMg^2+^	500	3

**Table 3 ijms-26-11027-t003:** Overview of simulation protocol.

	Equilibration				Production Run
Chronological step	1	2	3	4	5
Resolution	CG	CG	AA	AA	AA
Ensemble	NVT	NPT	NVT	NPT	NPT
Duration	2 ns	5 ns	0.5 ns	1 ns	500 ns
Timestep	10 fs	10 fs	2 fs	2 fs	2 fs
Position restraints	Protein/ligand/ structural ions	Protein/ligand/ structural ions	Protein/ligand/ structural ions	Protein/ligand/ structural ions	None
Thermostat	Berendsen	Velocity rescale	Nose–Hoover	Nose–Hoover	Nose–Hoover
Barostat	-	Berendsen	-	Parrinello–Rahman	Parrinello–Rahman

## Data Availability

The original contributions presented in this study are included in the article/Appendix A. Further inquiries can be directed to the corresponding author.

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
