# Peer review of "Derivatization of Bufadienolides at Carbon-3 of the Steroid Core and Their Consequences for the Interaction with Na^+^,K^+^-ATPase"

_ijms, 2025, doi:10.3390/ijms262211027_

Round 1

Reviewer 1 Report

Comments and Suggestions for Authors

The manuscript provides experimental and theoretical data on the interaction of a variety of bufadienolides with the Na+,K+-ATPase. Interestingly, the results show that the degree of glycosylation of the derivative has a strong effect on the binding affinity, with one glucose unit increasing binding affinity, but with further glucose units decreasing the affinity again. The MD simulations indicate that one glucose unit is favourable because of increased hydrogen bonding interactions.

I just have a small number of questions for the authors.

1) The experiments and simulations were carried out with the E2Pi conformation of the enzyme, i.e. the so-called back-door phosphorylated state. Why was this conformation chosen? Is it known that these derivatives interact with this conformation preferentially?

2) Experiments and simulations were also performed in the presence of 200 mM KCl. In the normal catalytic cycle, K+ would be expected to dephosphorylate the enzyme. Is this also the case for the E2Pi state, or is this resistant to dephosphorylation?

3) Figure 2C shows the affinity relative to anthroylouabain. Therefore, the y-axis is unitless. But how was the affinity of anthroylouabain quantified. Was it measured as a dissociation constant? This is what I assume, but I couldn't find it anywhere. This would explain the apparent inconsistency between Fig 2C and the text on the top of page 4, i.e., the figure shows a drop in relative affinity after adding one glucose, but the text on page 4 says that "attachment of a single sugar moiety significantly improves the affinity".

4) Minor typo: page 19, second to last line "enery" should be "energy".    

Author Response

We are thankful to both reviewers for their kind words and detailed and constructive comments, which have led to improvement of the manuscript. It is our hope that in such a form it is acceptable for publication.

The changes made are listed below and highlighted in the version Manuscript-marked-Ladefoged et al.docx uploaded as “non-published material”. 

Reviewer 1.

  • The experiments and simulations were carried out with the E2Pi conformation of the enzyme, i.e. the so-called back-door phosphorylated state. Why was this conformation chosen? Is it known that these derivatives interact with this conformation preferentially?
  • Experiments and simulations were also performed in the presence of 200 mM KCl. In the normal catalytic cycle, K+ would be expected to dephosphorylate the enzyme. Is this also the case for the E2Pi state, or is this resistant to dephosphorylation?

The following information and a new reference are added to the manuscript on p.3:

“Since affinity of CTS to Na+,K+-ATPase depends on the enzyme conformation and is highest towards its phosphorylated states, our choice fell on E2Pi phosphoform, a product of back-door phosphorylation with Pi as a substrate. It allowed to perform the binding experiments under equilibrium conditions (and not steady-state as in case of the ATPase reaction) and use the known crystal structures as start points for simulations. In addition, the back-door phosphorylation can occur in the presence of K+ [16] as visualized in the crystal structure of bufalin E2Pi complex obtained in the presence of 200 mM KCl [17].”

  • Figure 1C shows the affinity relative to anthroylouabain. Therefore, the y-axis is unitless. But how was the affinity of anthroylouabain quantified? Was it measured as a dissociation constant? This is what I assume, but I couldn't find it anywhere. This would explain the apparent inconsistency between Fig 1C and the text on the top of page 4, i.e., the figure shows a drop in relative affinity after adding one glucose, but the text on page 4 says that "attachment of a single sugar moiety significantly improves the affinity".

We have corrected the embarrassing error in the description  of y-axis in Fig. 1C and Fig.S1 and adjusted their legends. As pointed by the reviewer, the y-axis represents the Kd value for each compound relative to Kd for anthroylouabain (AO). Following information and a reference are added into Methods:

“The ratio of the residual AO fluorescence to the decrease in its fluorescence induced by the presence of another CTS (X), added in equal concentration, is proportional to (AO/KAO)/(X/KX) and reflects their relative affinities. For example, 50% decrease in AO fluorescence induced by a compound implies their equal affinities or equal equilibrium dissociation constants). The kinetics of interactions between AO and E2Pi under identical conditions was assessed in Fruergaard et al. [37]. Fig.2 in that paper shows: time-dependent changes in AO fluorescence as function of its concentration (binding velocities and amplitudes), calculated second-order association rate constants and a time course of AO dissociation from E2Pi. The absolute affinity was estimated to be higher than 0.1 nM.”

  • Minor typo: page 19, second to last line "enery" should be "energy". 

Done.

Reviewer 2 Report

Comments and Suggestions for Authors

Review of: ijms-3981046

This is really nice work that would be an excellent addition in IJMS. Thus, I recommend publication after the following issues have been addressed.

  • This is a nice computational modeling study that explore the structural features of the cardiac glycosides. My issue is primarily on of nomenclature. Particularly, what is meant by the N-glycoside as in “bufalin-N-glucose”. I can not find a depiction of this structure? Is it a “neo-glycoside” (i.e., NMeOMe replaces the linking oxygen)?
  • There are two long nomenclature traditions of using alpha-/beta- stereochemical labels, which in this context can be somewhat confusing. Would it be possible for the authors to also use the R/S-system as well. In this context, I believe the beta-bufalin, hellebrigenin, scillarenin, ouabagenin, digoxigenin and digitoxigenin structures depicted in Figure 1 are all shown with the C-3 A-ring alcohol in the S-configuration.
  • I think all the sugars investigated are beta-D-pyranosides. Is this true? Either way, can the structures be depicted?
  • It would be helpful for the readers if the stereochemistry at the A/B/C-ring fusion were also drawn.

Author Response

We are thankful to both reviewers for their kind words and detailed and constructive comments, which have led to improvement of the manuscript. It is our hope that in such a form it is acceptable for publication.

The changes made are listed below and highlighted in the version Manuscript-marked-Ladefoged et al.docx uploaded as “non-published material”. 

  • My issue is primarily one of nomenclature. Particularly, what is meant by the N-glycoside as in “bufalin-N-glucose”. I cannot find a depiction of this structure? Is it a “neo-glycoside” (i.e., NMeOMe replaces the linking oxygen)?

Yes, we have used N-glycoside and bufalin-N-glucose as synonyms for neo-glycoside and bufalin-MeON-glucose without explicitly stating this. The linker is shown in Figure 7 within the methods section, but this was too hidden. This point is now clarified.

line 156: “N-linked“ changed to “MeON-linked”

The new Supporting Figure S2 depicts structure of bufalin-N-glucose and serves as a reference throughout the manuscript.

  • There are two long nomenclature traditions of using alpha-/beta- stereochemical labels, which in this context can be somewhat confusing. Would it be possible for the authors to also use the R/S-system as well. In this context, I believe the beta-bufalin, hellebrigenin, scillarenin, ouabagenin, digoxigenin and digitoxigenin structures depicted in Figure 1 are all shown with the C-3 A-ring alcohol in the S-configuration.

We think the R/S-nomenclature is more intuitive for chemists, while alpha/beta is more common in biology, but ultimately agree that both can be present in the manuscript. We have therefore added R/S information to the new Supporting Figure S2. And yes, the C3-OH is S-configured.

  • I think all the sugars investigated are beta-D-pyranosides. Is this true? Either way, can the structures be depicted?

Yes, this is true. The structures have been included in the new Supporting Figure S2.

  • It would be helpful for the readers if the stereochemistry at the A/B/C-ring fusion were also drawn.

The stereochemistry of the core rings shown in Supporting Figure S2.